

# Identifying the critical state of cancers by single-sample Markov flow entropy

Juntan Liu[1], Yuan Tao[1], Ruoqi Lan[1], Jiayuan Zhong[1,2], Rui Liu[1] and Pei Chen[1,3]

[1] School of Mathematics, South China University of Technology, Guangzhou, Guangdong Province, China
[2] School of Mathematics and Big Data, Foshan University, Foshan, China
[3] Pazhou Lab, Guangzhou, Guangdong Province, China

## ABSTRACT

**Background:** The progression of complex diseases sometimes undergoes a drastic critical transition, at which the biological system abruptly shifts from a relatively healthy state (before-transition stage) to a disease state (after-transition stage). Searching for such a critical transition or critical state is crucial to provide timely and effective scientific treatment to patients. However, in most conditions where only a small sample size of clinical data is available, resulting in failure when detecting the critical states of complex diseases, particularly only single-sample data.

**Methods:** In this study, different from traditional methods that require multiple samples at each time, a model-free computational method, single-sample Markov flow entropy (sMFE), provides a solution to the identification problem of critical states/pre-disease states of complex diseases, solely based on a single-sample. Our proposed method was employed to characterize the dynamic changes of complex diseases from the perspective of network entropy.

**Results:** The proposed approach was verified by unmistakably identifying the critical state just before the occurrence of disease deterioration for four tumor datasets from The Cancer Genome Atlas (TCGA) database. In addition, two new prognostic biomarkers, optimistic sMFE (O-sMFE) and pessimistic sMFE (P-sMFE) biomarkers, were identified by our method and enable the prognosis evaluation of tumors.

**Conclusions:** The proposed method has shown its capability to accurately detect pre-disease states of four cancers and provide two novel prognostic biomarkers, O-sMFE and P-sMFE biomarkers, to facilitate the personalized prognosis of patients. This is a remarkable achievement that could have a major impact on the diagnosis and treatment of complex diseases.

# INTRODUCTION

The time evolution of a complex biological system is sometimes viewed as a time-dependent nonlinear dynamic system (*Gorban et al., 2021*; *Liu, Chen & Chen, 2020*), with the abrupt transition seen as the phase shift at a bifurcation point (*Scheffer et al.,*

Corresponding authors
Jiayuan Zhong,
201720127132@mail.scut.edu.cn
Pei Chen, chenpei@scut.edu.cn

*2009*). Therefore, from the viewpoint of dynamic systems, we can roughly model the progression of complex diseases (such as cancer, COVID-19, diabetes, *etc.*) as three distinct stages or states (Fig. 1A): (1) a before-transition state, a stable state with low fluctuation; (2) a critical/pre-disease state, an unstable state before the start of qualitative changes, high fluctuation (according to DNB theory (*Chen et al., 2012*), the variance of gene expression increases dramatically), sensitive to perturbation; (3) an after-transition/disease state, another stable state with low fluctuation after the qualitative transition. The pre-disease state is unstable and can be reverted to the before-transition state with appropriate intervention, while the after-transition/disease state is a stable state with high resilience and is nearly irreversible (*Chen et al., 2012*; *Liu et al., 2014*). Therefore, it is essential to identify the early warning signs of the critical transition during disease progression and recognize the pre-disease state, so that timely medical intervention can be administered to prevent or delay the progression of the disease.

Recently, the dynamic network biomarker (DNB) has been proposed to qualitatively describe the critical state of a biological system (*Chen et al., 2019*). Specifically, DNB theory uses differential equation theory to derive that systems exhibit three statistical properties when they reach the bifurcation point, allowing us to measure this dynamic difference at the molecular network level, not just gene expression. The proposal of DNB theory provides a basic framework for identifying the pre-disease states of complex diseases (*Chen et al., 2016*). For instance, successfully detect the pre-disease state of metabolic syndromes (*Koizumi et al., 2019*; *Liu et al., 2013*), and respiratory viral infections (*Gao et al., 2022*), identify immune checkpoint blockades (*Lesterhuis et al., 2017*), and assess cell fate commitment (*Peng et al., 2022*; *Zhong et al., 2022*). However, most developed methods are based on multiple samples, so multiple samples are required at each time, which limits their application in most real-world cases where only a single sample is available.

Specifically, inspired by the previous study (*Guo et al., 2021a*, *2021b*), this study proposes a model-free computational approach, single-sample Markov flow entropy (sMFE), which utilizes the inferred direct interaction network and the DNB concept to characterize the perturbation of a single sample, thus detecting the early-warning signal of critical transition during a complex disease process. Precisely, given reference samples from the normal cohort, sMFE is calculated for each single sample against the directed network, constructed based on rewiring the protein–protein interaction (PPI) network by gene networks inference using projection and lagged regression (GNIPLR) method (*Zhang, Chang & Liu, 2021*) (Fig. 1B). Two novel insights were provided in this computational method. On the one hand, by eliminating indirect interactions between genes, the established directed network more realistically reflects the molecular interactions in the biological network, and thus dynamic changes during complex disease progression can be more accurately captured. On the other hand, at the single-sample level, sMFE provides a reliable method for quantifying the pre-disease state or critical point of complex disease (Fig. 1C).

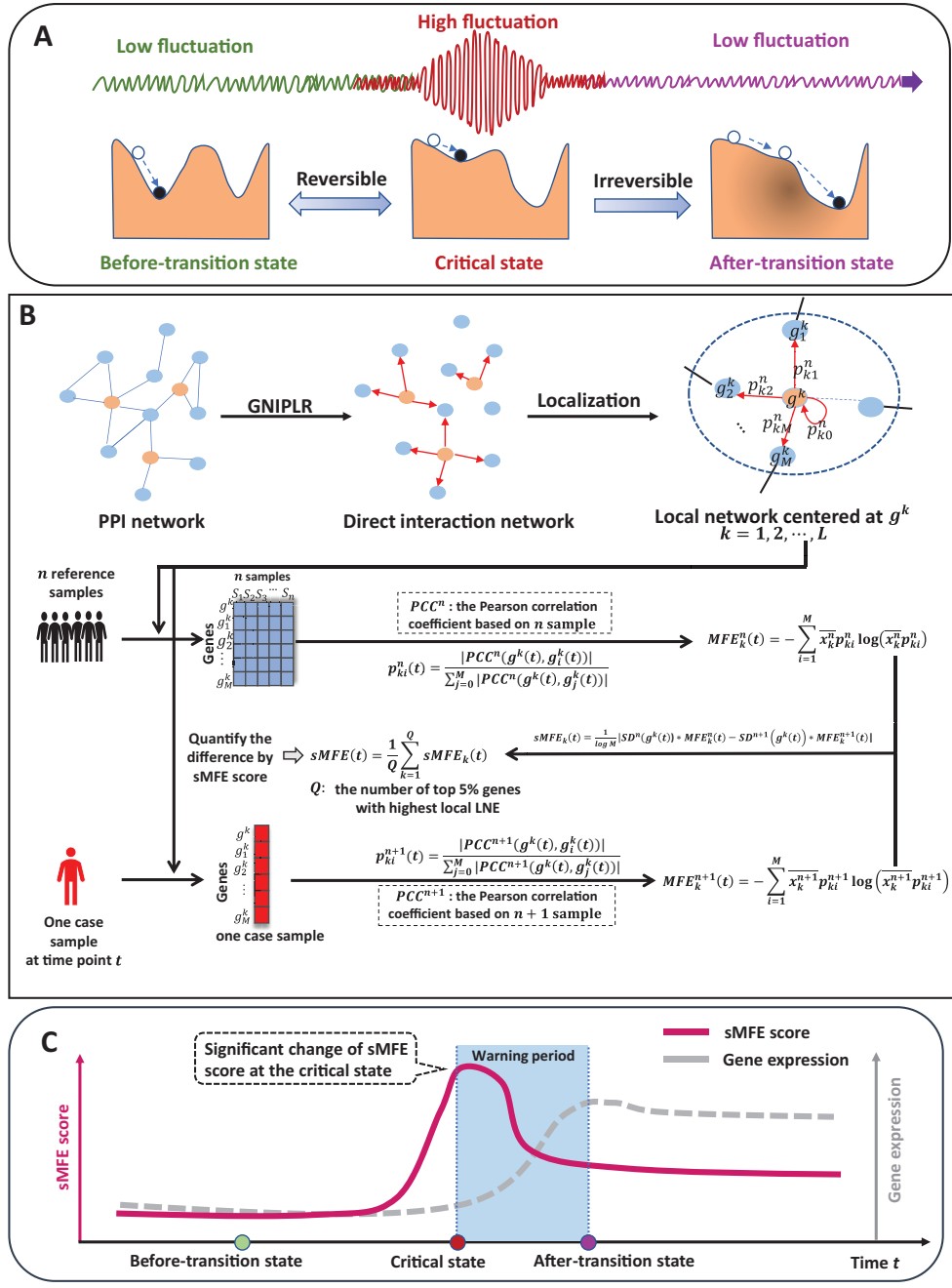

**Figure 1 The schematic of the sMFE method for detecting the critical state of complex diseases.** (A) The schematic diagram of the evolution of complex diseases. In dynamical systems theory, the evolution of disease can generally be divided into three stages or states, namely a relatively normal (before-transition) state, a pre-disease (critical) state, and a disease (after-transition) state. Among them, the relatively normal and the pre-disease states are reversible, while the disease state is irreversible. In addition, the expression of each molecular network in the before-transition and after-transition states is low-fluctuating and highly volatile in the critical state; (B) single-sample Markov flow entropy (sMFE) algorithm. Given a number of reference samples that can be obtained from normal cohort, and a directed network that used the GNIPLR (Gene networks inference using projection and lagged regression) method to orient the PPI network. Specifically, both the reference samples and the single-sample to be determined were mapped to this directed network, which can be divided into local networks. sMFE is calculated on a local network based on a single sample of any individual. For each local network centered on gene k, the local sMFE score is calculated; (C) during the evolution of complex diseases, the sMFE score (Eq. (7)) suddenly increases in the pre-disease state. This mutation in the sMFE indicates a complex biological system's tipping point (or critical state).

To test the effectiveness of sMFE, four different tumor datasets from TCGA, including esophageal carcinoma (ESCA), colon adenocarcinoma (COAD), kidney renal clear cell carcinoma (KIRC), lung adenocarcinoma (LUAD), were used to validate our algorithm. In all datasets, the critical states identified by sMFE preceded severe disease deterioration, *i.e.*, a significant alteration in sMFE score indicated an early warning of the critical transition to disease deterioration. By searching the literature, we found that the following clinical characteristics during the deterioration of diseases (*e.g.*, cancer) occur, and distant metastasis occurs in the later stage of cancer development, which is difficult to recover by external treatment and can be regarded as the basis for disease deterioration. Specifically, the critical state for ESCA was identified in stage IIIA disease before distant metastasis, and stage IIB was identified as the critical state for COAD before lymph node metastasis; stage II was identified critical state for KIRC before the tumor migrates to form distant metastasis, and that of LUAD was identified in stage IIIB disease before the tumor cells invaded distant tissues or organs. In addition, we further analyze the signaling gene and propose two types of biomarkers, *i.e.*, optimistic sMFE (O-sMFE) and pessimistic sMFE (P-sMFE) biomarkers (see Fig. S1 for the specific procedure), combined with survival analysis. Statistically, O-sMFE biomarkers were associated with good prognosis, while P-sMFE biomarkers were linked to poor prognosis. Furthermore, our results were validated by functional and pathway enrichment analyses.

## MATERIALS AND METHODS

### Theoretical background

The DNB theory (*Chen et al., 2012*; *Liu, Aihara & Chen, 2013*) is proposed to visualize the evolution of complex diseases into three periods: a normal (before-transition) state, a pre-disease (critical) state, and a disease (after-transition) state. The normal state is stable and resilient to perturbations, while the pre-disease state is unstable and reversible with appropriate intervention. The disease state is also stable and resilient, but is almost irreversible (*Litt et al., 2001*; *Liu, Chen & Chen, 2020*; *McSharry, Smith & Tarassenko, 2003*; *Venegas et al., 2005*). Moreover, when a complex system is near the critical point, a dominant group of DNB biomolecules can be identified, based on observed data, that satisfy the following three conditions (*Chen et al., 2012*a):

- The Pearson correlation coefficient ($PCC_{in}$) between any pair of members in the DNB group rapidly increases;
- The Pearson correlation coefficient ($PCC_{out}$) between one member of the DNB group and any other non-DNB member rapidly decreases;
- The standard deviation ($SD_{in}$) or coefficient of variation for any member in the DNB group drastically increases.

Our DNB theory suggests that phase transitions are characterized by a set of highly fluctuating and highly correlated features/variables, which implies an imminent transition

to the disease state. Therefore, we can quantify the tipping points based on these three conditions, provide early-warning signals of disease deterioration, and then determine the biomolecular dominant group to make up DNB members. These three conditions are the basis of DNB theory and have many applications in disease progression and biological processes to predict its critical state (*Liu, Aihara & Chen, 2013*; *Liu, Chen & Chen, 2020*).

Markov chain entropy (MCE) (*Shi et al., 2018*) was constructed to represent the potential of samples containing scRNA-Seq or bulk RNA-Seq information. This method requires the normalized RNA Seq profile and a defined interaction network between the genes.

Specifically, given an interaction network G with $n_v$ nodes and $n_E$ edges, and the transition probability matrix $P = \{p_{ij}\}$, where $i, j = 1, \ldots, n_E$. Where $p_{ij} = 0$ if there is no edge between two different nodes i and j. Then the Markov chain entropy based on network G is defined as

$$\text{MCE} = -\sum_{(i,j)\in \bar{E}} \pi_i p_{ij} \log\left(\pi_i p_{ij}\right) \tag{1}$$

where $\pi_i$ was node i's expression normalized to [0,1]. In this article, MCE was split into each node to form Markov flow entropy (MFE) (Fig. 1B). For centered gene $g^k$, the MFE is defined as

$$\text{MFE}(g^k) = -\sum_{j=1}^{M} \pi_i p_{ij} \log\left(\pi_i p_{ij}\right) \tag{2}$$

where M denotes the number of the 1st-order neighbors $(g_1^k, \ldots, g_M^k)$ of $g^k$, $p_{ij}$ denotes the transition probability from $g^k$ to $g_j^k$, and $\pi_i p_{ij}$ means the probability transmitted from $g^k$ to $g_j^k$ along the directed edge.

## Algorithm to identify the tipping point based on sMFE

Reference samples (healthy or relatively healthy cells; tumor-adjacent samples here) and only one disease sample were required in the sMFE method to identify the tipping point with the following algorithm (Fig. 1B).

(Step 1) Constructing a global directed network $N^G$ by mapping the genes to a protein–protein interaction (PPI) network from STRING (http://string-db.org) (*Szklarczyk et al., 2015*), with a confidence level of 0.800. Discard isolated nodes that do not have any links with other nodes. The direct interaction network was constructed based on rewiring the protein–protein interaction (PPI) network by GNIPLR (gene networks inference using projection and lagged regression) (*Zhang, Chang & Liu, 2021*) method. Tumor-adjacent samples were used to infer the gene regulatory network.

(Step 2) Mapping data into the global network $N^G$ constructed in the previous step to form expression matrix. Mapping datasets of four cancers downloaded from the TCGA database to the global network $N^G$, then the gene expression matrix formed for further analysis.

(Step 3) Calculating the local Markov flow entropy (MFE) for each gene. Specifically, we extract each gene $g^k$'s local network $N^k$ (k = 1, 2, ..., L) from the global network $N^G$ (Fig. 1B), the number of the local network is $L$, and $\{g_1^k, \ldots, g_M^k\}$ are the 1st-order neighbors of $g^k$. Then, based on n reference samples, the formula for calculating local Markov flow entropy $MFE_k^n(t)$ is

$$MFE_k^n(t) = -\sum_{i=1}^{M} \overline{x_k^n} p_{ki}^n \log\left(\overline{x_k^n} p_{ki}^n\right), \tag{3}$$

with

$$p_{ki}^n(t) = \frac{\left|PCC^n\left(g_i^k(t), g^k(t)\right)\right|}{\sum_{j=0}^{M} \left|PCC^n\left(g_j^k(t), g^k(t)\right)\right|}, \tag{4}$$

where the constant $M$ represents the count of neighbors in the local network $N^k$ and $PCC^n\left(g_i^k(t), g^k(t)\right)$ represents the Pearson correlation coefficient (PCC) between the center gene $g^k$ and its 1-st neighbor $g_i^k$ based on $n$ reference samples at time point $t$. Specifically, $g_i^k = g^k$ if $j = 0$. $\overline{x_k^n}$ is the mean of the canter gene $g^k$'s normalized expression based on $n$ reference samples at time point $t$.

(Step 4) Calculating the differential MFE called $sMFE_k(t)$. Under $n + 1$ mixed samples, where a single sample from an individual was added to $n$ reference samples at time point $t$, the same with the prior step, $MFE_k^{n+1}(t)$ is calculated, i.e.,

$$MFE_k^{n+1}(t) = -\sum_{i=1}^{M} \overline{x_k^{n+1}} p_{ki}^{n+1} \log\left(\overline{x_k^{n+1}} p_{ki}^{n+1}\right) \tag{5}$$

Then, the $sMFE_k(t)$ is calculated as

$$sMFE_k(t) = \frac{1}{\log M}\left|SD^n\left(g^k(t)\right) * MFE_k^{nt}(t) - SD^{n+1}\left(g^k(t)\right) + MFE_k^{n+1}(t)\right|, \tag{6}$$

where $SD^n(g^k(t))$ and $SD^{n+1}(g^k(t))$ represent the standard deviation of the gene expression for the center gene $g^k$ based on $n$ reference samples and $n + 1$ mixed samples at time point $t$, respectively.

(Step 5) The fifth step is to calculate the global single-sample Markov flow entropy $sMFE(t)$, i.e.,

$$sMFE(t) = \frac{1}{Q}\sum_{k=1}^{Q} sMFE_k(t), \tag{7}$$

where constant $Q$ represents the count of top 5% genes with highest local $sMFE$. Furthermore, t-test was applied to test if the identified critical state significantly different with the other states. The alternative state was viewed as critical state if $P < 0.05$ (see Supplemental Information). From the formula of sMFE, we can see that when the interaction between the center node and the 1-st neighbor node converges (that is, the transition probability is the same), the sMFE value reaches the maximum value.

Conversely, when the interaction between the center node and one 1-st neighbor node is extremely strong (the transition probability equals 1), the sMFE value reaches the minimum. From DNB theory, it can know that in the sample at the critical point period, the interaction between DNB molecules and their neighboring nodes will increase extremely so that it can form a significant difference from the reference sample, and sMFE can measure this difference, and when the sample reaches the critical point, the significance increases so that it can warn the disease evolution process. According to DNB theory (*Chen et al., 2012*; *Liu, Aihara & Chen, 2013*), when the system approaches the critical point, the sMFE score can effectively characterize the fluctuation of the network, and thus can be regarded as an early warning signal prior to critical deterioration.

## Data processing and functional analysis

The Cancer Genome Atlas (TCGA) database (GDC (cancer.gov)) was used to download four unrelated clinical tumor datasets: esophageal carcinoma (ESCA), colon adenocarcinoma (COAD), (kidney renal clear cell carcinoma (KIRC)), and lung adenocarcinoma (LUAD). RNA-seq data from tumor and tumor-adjacent samples, as well as clinical information, were included in these datasets. Then we classified the tumor samples into different stages based on clinical (stage) information downloaded from TCGA, samples lacking stage information were discarded, and Table S1 provided the full clinical staging information.

The molecular global template network was constructed using the steps shown in Fig. 1A.

The first step in constructing the molecular global template network was to download the protein–protein interaction networks for *Homo sapiens* from STRING (http://string-db.org). This data was then used to generate a global template network, which was composed of nodes representing proteins and edges representing interactions between proteins. All interaction used for discussion was picked with high confidence (higher than 0.8). Nodes that are not connected to any other nodes are removed from the PPI network.

Second, the direct interaction network was constructed based on rewiring the protein–protein interaction (PPI) network by GNIPLR (gene networks inference using projection and lagged regression) (*Zhang, Chang & Liu, 2021*) method. Tumor-adjacent samples were used to infer the gene regulatory network (GRN; directed network) for subsequent analysis (detailed information for how to choose adjustable parameters in GNIPLR were added in Supplemental Information).

In the end, the directed network, created in prior step, was adapted to map the gene expression of each RNA-seq dataset to form a molecular network for further analysis.

The Kyoto Encyclopedia of Genes and Genomes (KEGG) database Mapper tool (KEGG Mapper Color) and the Database for Annotation, Visualization and Integrated Discovery (DAVID) Functional Annotation Tool (DAVID: Functional Annotation Tools (ncifcrf.gov)) were used to conduct the enrichment analysis. These statistical analysis carried in the article was done using R Software v 4.0.3 (R: The R Project for Statistical Computing
(r-project.org)), and the R Survival package was used to make Cox survival analysis. The visualization results and network analysis are implemented using Cytoscape software (www.cytoscape.org).

# RESULTS

## Identifying the critical transition points of four cancers with sMFE

The sMFE algorithm was used to identify the critical transition points of four cancers (ESCA, COAD, KIRC, and LUAD) from TCGA datasets. These datasets contain much clinical information, in which doctors divide cancer patients into different stages according to the size of the tumor diameter, such as KIRC samples were divided into four stages (I, II, III, and IV), while LUAD samples were divided into seven stages (IA, IB, IIA, IIB, IIIA, IIIB, and IV). The tumor-adjacent (TA) samples were deemed relatively normal and used as a reference for each cancer when calculating sMFE according to the proposed algorithm. The average sMFE score curve for each cancer stage was calculated to reflect the dynamic change and was shown in Figs. 2A–2D. Based on DNB theory, the critical state during disease progression can be reflected by the tipping point of the sMFE score curve (Fig. 1A). Kaplan–Meier (log-rank) survival analysis was performed on the prognosis of before-transition and after-transition samples (Figs. 2E–2H) to further validate the identified critical state.

As seen in Fig. 2A (yellow curve), the average sMFE score of ESCA increased significantly ($P = 0.0077$) at stage IIIA, suggesting an upcoming critical transition after stage IIIA.; that is, the tumor migrates and invades to form distant metastasis at stage IIIB and ultimately cause distant metastasis (stage IV) (*Dong et al., 2022*). While at the six-time points, there was little significant difference among the average gene expression of differentially expressed genes (DEGs) (the gray curve in Fig. 2A). Of note, samples taken before and after the transition point (Stage IIIA) had significantly different survival curves ($P < 0.0001$; Fig. 2E), and patients taken after the transition point (Stage IIIA) had shorter survival times than those before the transition.

When applied to COAD, a significant change ($P = 0.0045$; Fig. 2B) in sMFE was detected around stage IIB, suggesting lymph node metastasis and tumor invasion of other adjacent organs in stage III (*Hari et al., 2013*). As presented in gray in Fig. 2B, dynamic changes ($P = 0.16$) in the mean gene expression of DEGs were detected in stage IIIB and therefore failed to provide timely early-warning signals of the critical transition. Moreover, it is worth noting that survival analysis showed that the survival time of samples taken before the transition point (Stage IIB) was significantly different than that taken after the transition point (Stage IIB) ($P < 0.0001$; Fig. 2F). Furthermore, patients from after the identified critical state (Stage IIB) had shorter survival times than those from before the transition.

For KIRC dataset, a drastic increase ($P = 0.014$) in the sMFE score was observed in stage II (transition point; Fig. 2C), indicating an upcoming critical transition into the disease state (stage III/IV), where the tumor migrates to form distant metastasis at stage III (*Gu & Zhao, 2019*). In terms of mean gene expression, dynamic changes ($P = 0.25$; the gray curve in Fig. 2C) were detected in stage III and therefore failed to provide timely early-warning

signals of the critical transition. Furthermore, survival analysis demonstrated that survival curves had a significant difference ($P < 0.0001$) between stage I~II and stage III~IV KIRC samples according to clinical information (Fig. 2G). In addition, the survival time of patients after stage II was significantly shorter than before stage II.

The red curve presented the LUAD data in Fig. 2D, the drastic increase of sMFE occurred at stage IIIB ($P = 0.0027$), indicating an upcoming critical transition after stage IIIB; that is, stage IV was characterized by a distant metastasis process, in which the tumor cells invaded distant tissues or organs (*Chiang & Massagué, 2008*). However, the gray curve in Fig. 2D shows little significant increase in the mean expression of DEGs occurring among the six-time points. Of note, the survival curves of samples taken before and after the transition point (stage IIIB) tend to show a significantly different ($P = 0.0024$; Fig. 2H). In addition, patients with stage after stage IIIB had shorter survival times than those before. As shown, sMFE is an effective approach to identify the critical state of cancers.

## The dynamic evolution of gene regulatory networks

To validate the critical states identified by sMFE, we further examined them from the network perspective. First, for each sample, the top 5% of genes with the highest sMFE scores were considered the signaling genes. Dynamic network biomarkers (DNBs) consist of common signaling genes in samples in the identified critical state. In DNB theory, these DNB molecules exhibit statistical properties when the system reaches a critical point. Biologically, they may be involved in important biological reaction processes during disease progression. These DNB molecules were mapped to the STRING database to form a connected PPI network, and their dynamic changes in the process of disease evolution were studied from the network level.

The landscape of the local sMFE score for ESCA data was depicted in Fig. 3A. It was worth noting that the local sMFE scores of the signaling genes showed an abruptly increased collaborative manner around stage IIIA. Furthermore, in the network structure, a noticeable change occurred at stage IIIA (Fig. 3E), indicating an impending critical transition (*Dong et al., 2022*), which spontaneously come together with the experimental results. When applied to the COAD dataset, the landscape of the local sMFE score was shown in Fig. 3B. It was discovered that the peak of local sMFE scores for signaling genes appeared at stage IIB. Furthermore, there was a significant change in the network structure at stage IIB (Fig. 3F), indicating lymph node metastasis and tumor invasion of other adjacent organs in stage III (*Hari et al., 2013*). As shown in Fig. 3C, for KIRC, showing that the peak local sMFE of signaling genes appeared at stage II. Moreover, in the PPI network, a drastic change occurred in stage II (Fig. 3G), where the tumor gradually migrates to form distant metastasis at stage III (*Gu & Zhao, 2019*). Results for LUAD in Fig. 3D, the local sMFE score landscape peaked in stage IIIB and there was a significant change in the network structure of stage IIIB (Fig. 3H), *i.e.*, stage IV was characterized by a distant metastasis process, in which the tumor cells invaded distant tissues or organs (*Chiang & Massagué, 2008*). In addition, Fig. S2 provided results of the dynamic network changes across all stages.

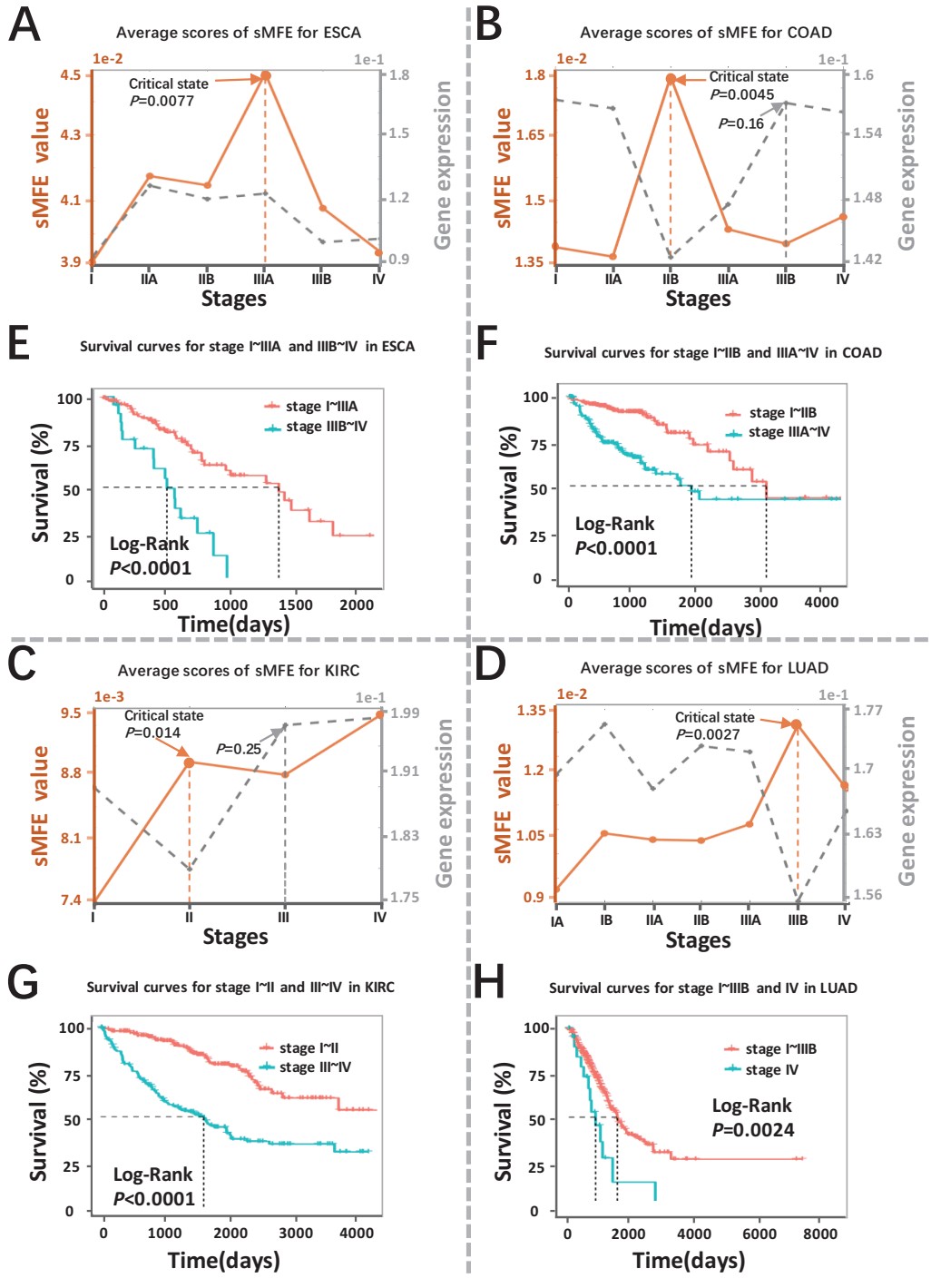

**Figure 2** **Detection of the critical states for the four cancers.** Dynamic changes between sMFE and the mean DEGs gene expression across four cancer datasets: (A) ESCA; (B) COAD; (C) KIRC; (D) LUAD. Survival analysis for samples adopt from before and after the critical state for four cancers: (E) ESCA; (F) COAD; (G) KIRC; (H) LUAD.

## Prognostic biomarkers of tumors in the sMFE method

In order to identify biomarkers that were effective in predicting prognoses. Indeed, signaling genes could be categorized into two types of molecules for prognostic prediction as common biomarkers for all samples: those for samples with poor prognosis, termed pessimistic sMFE (P-sMFE) biomarkers; and those with good prognosis, termed optimistic sMFE (O-sMFE) biomarkers. Furthermore, those two types of biomarkers are not only the signaling genes, pessimistic/optimistic biomarkers, but also the non-differentially expressed genes usually screened out by most researchers in the first step. More details about the identification method are provided in Fig. S1.

The meaning of these two biomarkers can also be understood in this way, samples of optimistic sMFE biomarkers that appeared in their signaling gene had better-predicted prognoses than those without; in other words, these samples were expected to survive longer than others. Conversely, samples with pessimistic sMFE biomarkers in their signaling genes had lower predicted prognoses than those without; that is, these samples were expected to survive for a shorter time than other samples. A total of four genes were identified as optimistic sMFE biomarkers for ESCA (Fig. 4A and Table S2), while six genes were identified as pessimistic sMFE biomarkers (Fig. 4A and Table S2). Specifically, survival times of samples with O-sMFE biomarkers *SCYL3* and *TRPS1* ($P = 0.016$ and 0.036 respectively) were longer than those without, as shown in Fig. 4A. Conversely, samples with P-sMFE biomarkers *EPCAM* and *POLK* ($P = 0.0057$ and 0.0064, respectively) had shorter survival times than those without.

In the analysis for COAD, *AGR2* ($P = 0.044$; Fig. 4B) and *PCM1* ($P = 0.046$; Fig. 4B) were identified as O-SMFE biomarkers, and it's seen the survival time of these identified samples tend to be longer than that of other samples according to survival curve analysis (Fig. 4B). Additionally, *ATP1A3* ($P < 0.0001$; Fig. 4B) and *PARL* ($P = 0.00093$; Fig. 4B) were identified as P-SMFE biomarkers, survival analysis in Fig. 4B showed the survival time of these identified samples tend to be shorter than that of other samples. When applied to KIRC dataset, samples with O-sMFE biomarkers *MAML2* and *RNF5* ($P = 0.0058$ and 0.0034, respectively) had better prognoses, with longer survival times, compared to other samples. Conversely, samples with P-sMFE biomarkers *BST1* and *VPS39* ($P = 0.0085$ and 0.00011, respectively) had worse prognoses, with shorter median survival times of almost 5 years. As shown in Fig. 4D, for LUAD, *ACOX2* ($P = 0.021$; Fig. 4D) and *KHDRBS3* ($P = 0.027$; Fig. 4D) were identified as O-SMFE biomarkers and *FLNC* ($P = 0.0021$; Fig. 4D) and *FZD1* ($P = 0.0091$; Fig. 4D) were identified as P-SMFE biomarkers.

We propose a method to uncover novel biomarkers, drug targets and key regulators, which are usually overlooked by traditional studies due to their non-differential expression. In addition, all identified optimistic/pessimistic sMFE biomarkers for the four cancers were supplemented in Table S2.

## Functional analysis of common signaling genes

Common signaling genes for the four cancers were extracted for functional analysis (Fig. 5A). As shown in Fig. 5B, not only were there occurred rich intersections among signaling genes from different cancers, but their biological functions were also closely

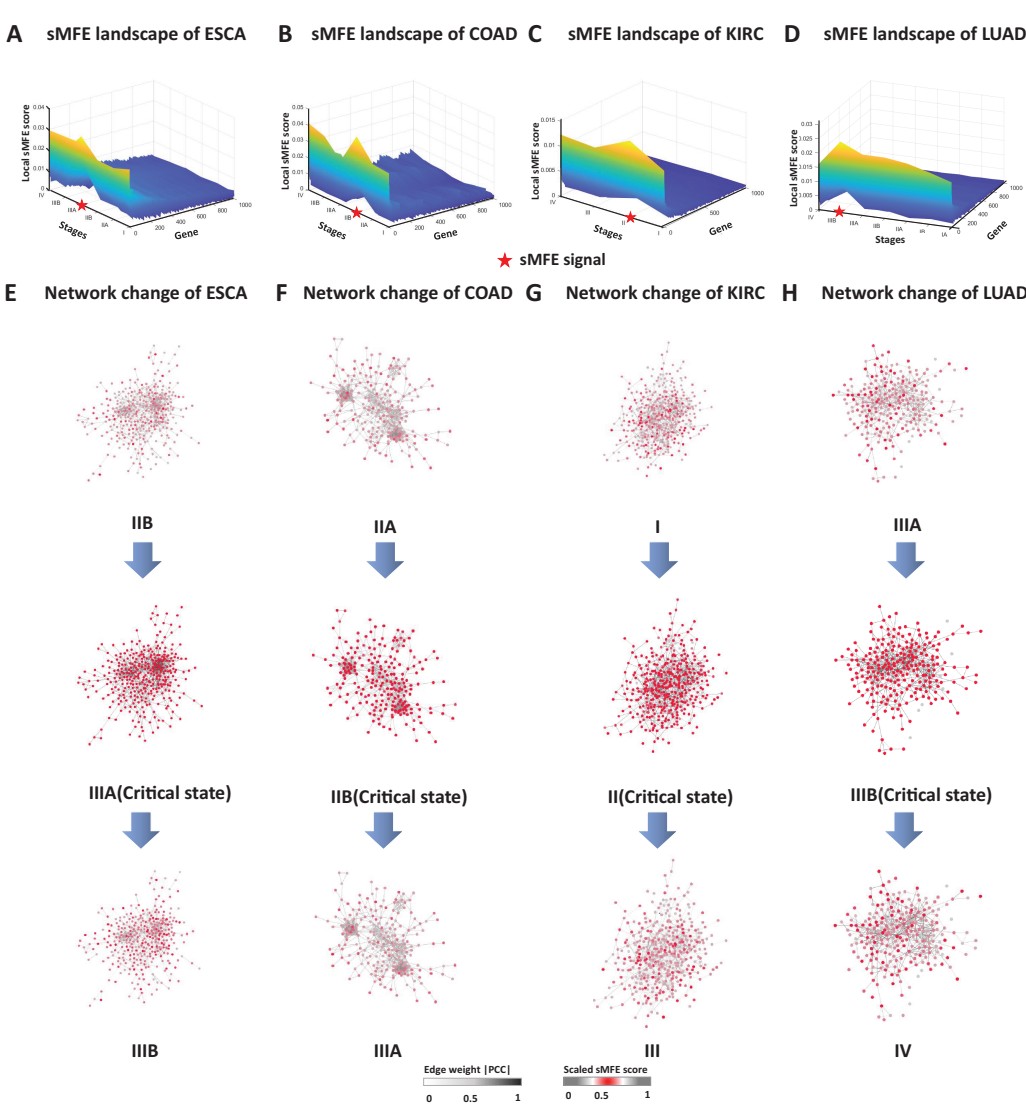

**Figure 3 The dynamic evolution of the regulatory networks for signaling genes in ESCA, COAD, KIRC, and LUAD.** The landscape of local sMFE values illustrates the dynamic evolution of network entropy in a global view for ESCA (A), COAD (B), KIRC (C), and LUAD (D), respectively. Key gene regulatory networks were reconstructed using sMFE for the signaling genes (top 5% genes with the largest local sMFE value). The network structure was derived by mapping s ignaling genes to the STRING PPI network. We discarded all the isolated nodes without any links to other nodes. The color of each node represents the mean local sMFE value, and the color of each edge represents the absolute value of the Pearson correlation coefficient. (A) In ESCA, the signaling gene group/module evolved, and there was a significant change in the network structure at stage IIIA. (B) Similarly, the network sMFE score significantly changed at stage IIB in COAD. (C) The network sMFE score significantly changed at stage II in KIRC. (D) The network sMFE score significantly changed at stage IIIB in LUAD.

related. In addition, we performed the KEGG pathway enrichment analysis of common signaling genes, where their samples located in tipping points, to explore the biological processes in which they participated. For these four tumor datasets, as shown in Figs. 5C–5F, these signaling genes were significantly associated with cancer-associated pathways. For example the cell cycle (*Hartwell & Kastan, 1994*), PI3K-Akt signaling

pathway (*Martini et al., 2014*), spliceosome (*Yang, Beutler & Zhang, 2022*), Wnt signaling pathway (*Zhou et al., 2022*). In addition, Figs. 5G and 5H illustrated that common signaling genes were mainly enriched in the above-mentioned pathway, which suggests that these common signaling genes may serve an important contributor to cancer development and progression. For instance, *Ras-related C3 botulinum toxin substrate 2 (RAC2)* promotes abnormal proliferation of quiescent cells by enhanced *JUNB* expression *via* the *MAL-SRF* pathway (*Pei et al., 2018*), *PIK3CA*-mutation was reported related to metastatic breast cancer (*Higgins et al., 2012*; *Mosele et al., 2020*), *Protein tyrosine kinase 2 (PTK2)* was a novel therapeutic target to overcome acquired *EGFR-TKI* resistance in non-small cell lung cancer and as a driver of radioresistance in HPV-negative head and neck cancerPTK2/FAK and radioresistance (*Skinner et al., 2016*; *Tong et al., 2019*), *RhoA* is associated with invasion and poor prognosis in colorectal cancer (*Jeong et al., 2016*).

## DISCUSSION

It is crucial to hunt for the critical state of complex biological systems. Suppose an early warning signal of the critical state can be provided before the disease deterioration. In that case, appropriate time can be provided to prevent or at least prepare for a catastrophic deterioration. However, small sample sizes are a challenge in biological studies and clinical practice, especially in cancer, and often lead to model errors and biases in the analysis. To overcome these problems when identifying transition points or critical states that precede disease states, the single-sample Markov flow entropy (sMFE) method was proposed in this study. Its effectiveness and reliability have been tested across multiple diseases.

In this study, we developed a computational method, single-sample Markov flow entropy (sMFE), to explore the dynamic changes in combined effects on molecular associations and thus characterize the perturbation of a single sample close to the tipping point. In addition, functional enrichment analysis revealed that the common signaling genes for four cancer datasets are involved in significant biological processes or pathways, for instance, the cell cycle (*Hartwell & Kastan, 1994*), PI3K-Akt signaling pathway (*Martini et al., 2014*), spliceosome (*Yang, Beutler & Zhang, 2022*), Wnt signaling pathway (*Zhou et al., 2022*) (Fig. 5). Further analysis revealed two distinct types of biomarkers derived from the signaling genes (*Liu et al., 2019*), *i.e.*, pessimistic sMFE (P-sMFE) and optimistic sMFE (O-sMfE) biomarkers. Furthermore, those two types of biomarkers not only the signaling genes, pessimistic/optimistic biomarkers, but also the non-differentially expressed genes and are therefore usually screened out by most researchers in the first step, so they can provide novel insight for further study into the molecular mechanisms of how tumor onset and disease deterioration work.

In a word, the sMFE approach has several important advantages. First, this method is a model-free data-driven and single-sample-based algorithm, which is conducive to the development of personalized medicine. Second, our method was based on the direct interaction network, constructed from an *a priori* knowledge-based PPI network by GNIPLR, which is different from the past studies that mainly focused on undirected networks. Third, the sMFE method provided two distinct types of biomarkers applicable

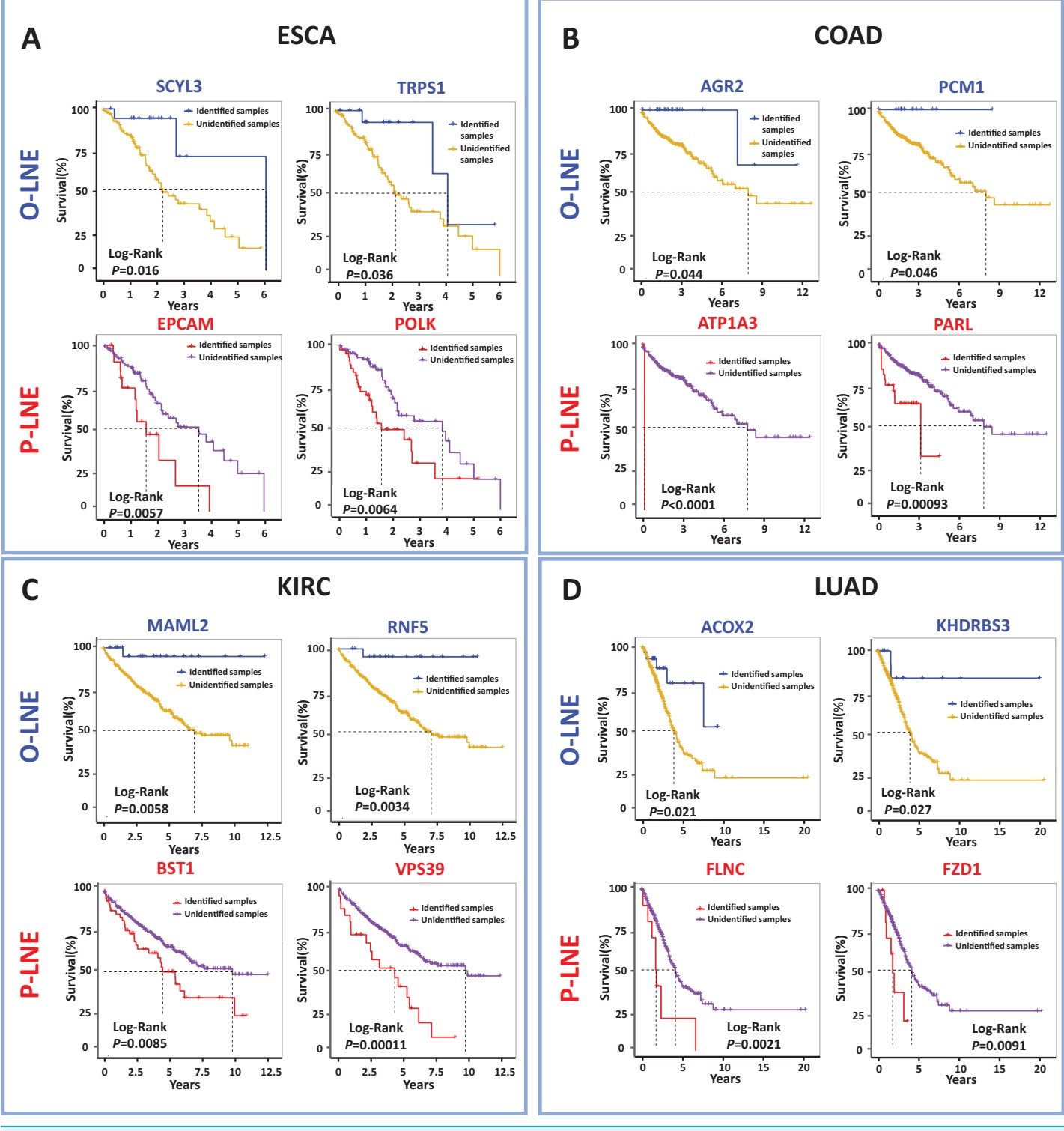

**Figure 4 Comparison of survival curves between samples with and without O-sMFE and P-sMFE biomarkers for ESCA, COAD, KIRC, and LUAD.** (A) The O-sMFE biomarkers *SCYL3* and *TRPS1* and the P-sMFE biomarkers *EPCAM* and *POLK* in ESCA; (B) the O-sMFE biomarkers *AGR2* and *PCM1* and the P-sMFE biomarkers *ATP1A3* and *PARL* in COAD; (C) the O-sMFE biomarkers *MAML2* and *RNF5* and the P-sMFE biomarkers *BST1* and *VPS39* in KIRC; (D) the O-sMFE biomarkers *ACOX2* and *KHDRBS3* and the P-sMFE biomarkers *FLNC* and *FZD1* in LUAD. Samples with O-sMFE or P-sMFE biomarkers in their signaling genes were deemed samples with biomarkers, while others were deemed samples without biomarkers.

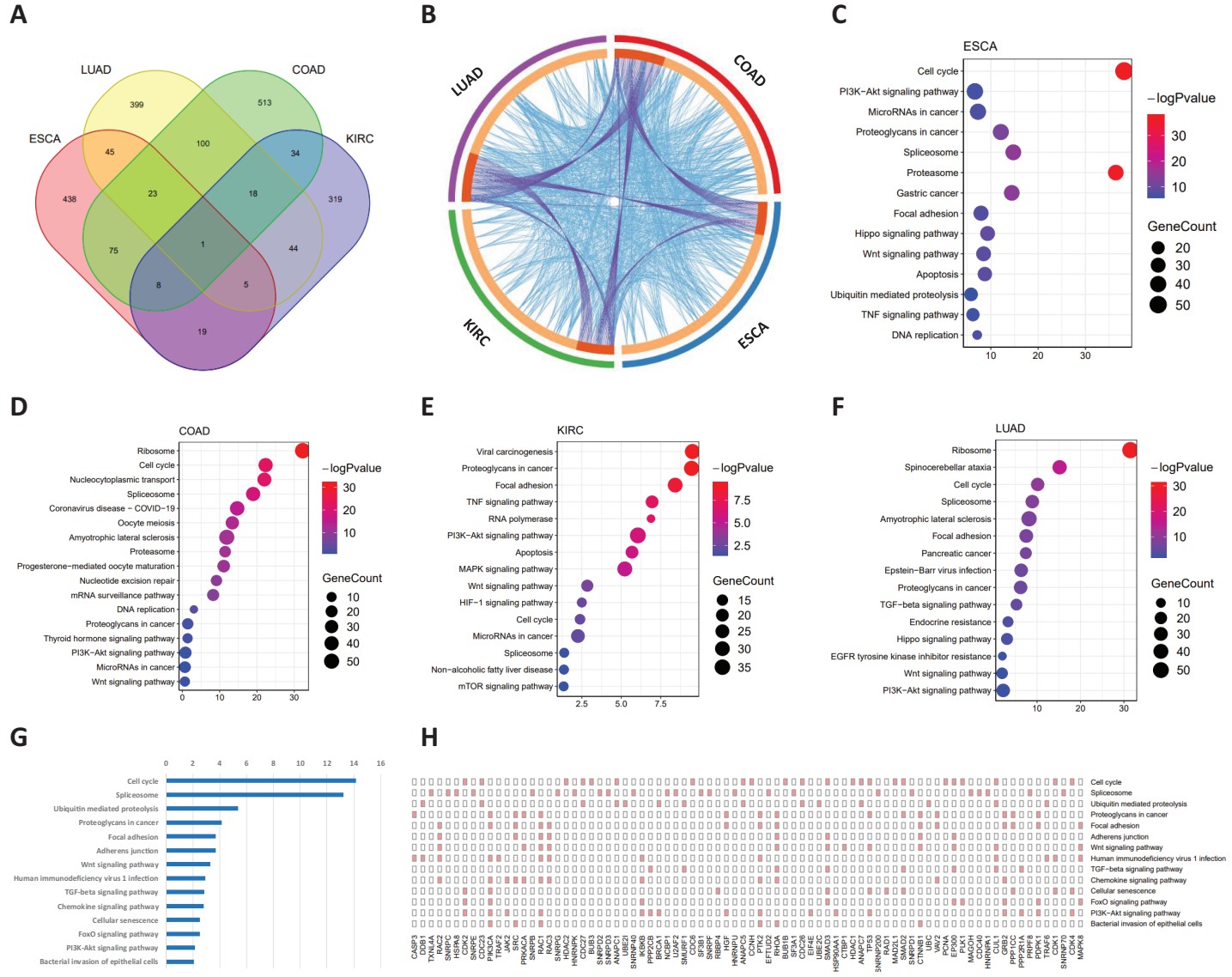

**Figure 5 Functional analysis of common signaling genes in different cancer datasets.** (A) Common signaling genes among ESCA, LUAD, COAD, and KIRC. (B) There is a considerable overlap of the same signaling genes and their biological functions among the four cancers. The outer ring represents different cancer-signaling genomes, and the inner ring represents identical genes and functions. The interconnections of identical genes are indicated by purple lines and functional links by blue lines. KEGG pathway enrichment analysis of signaling genes in four cancers: ESCA (C), COAD (D), KIRC (E), and LUAD (F). (G) Common signaling genes are involved in multiple cancer-related pathways. (H) Detailed common signaling genes in cancer-related pathways.

for identifying novel biomarkers unlike any previous one, drug targets in drug development and prognostic indicators for prognostic analysis.

# CONCLUSIONS

The sMFE method provides a solution to the identification problem of critical states/pre-disease states of complex diseases solely based on a single sample, making it suitable for real-world clinical data. In addition to effectively identifying the critical states or transition points of the four cancers, our method provided two new prognostic biomarkers,

optimistic sMFE (O-sMFE) and pessimistic sMFE (P-sMFE) biomarkers. Therefore, this method has great practical application potential in the fields of personalized medicine, identification of molecular mechanisms of disease progression, and preventive medicine.

## LIST OF ABBREVIATIONS

| | |
|---|---|
| **sMFE** | Single-sample Markov flow entropy |
| **DNB** | The Dynamical Network Biomarker |
| **LUAD** | Lung adenocarcinoma |
| **COAD** | Colon adenocarcinoma |
| **KIRC** | Kidney renal clear cell carcinoma |
| **ESCA** | Esophageal carcinoma |
| **TCGA** | The Cancer Genome Atlas |
| **SD** | Standard deviation |
| **PCC** | Pearson correlation coefficient |
| **signaling genes** | The top 5% of genes with the highest sMFE score |
| **O-sMFE biomarker** | Optimistic sMFE biomarker |
| **P-sMFE biomarker** | Pessimistic sMFE biomarker |
| **DEGs** | Differentially expressed genes |

### Funding

This work was supported by the National Natural Science Foundation of China (Nos. 62172164, 12026608, and 11971176) and the Guangdong Basic and Applied Basic Research Foundation (Grant Nos. 2019B151502062, 2021A1515012317). The funders had no role in study design, data collection and analysis, decision to publish, or preparation of the manuscript.

### Grant Disclosures

The following grant information was disclosed by the authors:
National Natural Science Foundation of China: 62172164, 12026608, and 11971176.
Guangdong Basic and Applied Basic Research Foundation: 2019B151502062, 2021A1515012317.

### Competing Interests

The authors declare that they have no competing interests. Pei Chen is employed by Pazhou Lab.

### Author Contributions

- Juntan Liu performed the experiments, analyzed the data, prepared figures and/or tables, authored or reviewed drafts of the article, and approved the final draft.
- Yuan Tao analyzed the data, authored or reviewed drafts of the article, and approved the final draft.

- Ruoqi Lan analyzed the data, authored or reviewed drafts of the article, and approved the final draft.
- Jiayuan Zhong conceived and designed the experiments, analyzed the data, authored or reviewed drafts of the article, and approved the final draft.
- Rui Liu conceived and designed the experiments, authored or reviewed drafts of the article, and approved the final draft.
- Pei Chen conceived and designed the experiments, authored or reviewed drafts of the article, and approved the final draft.

## Data Availability

The source code of algorithm is available in the Supplemental File.

## Supplemental Information

Supplemental information for this article can be found online at http://dx.doi.org/10.7717/peerj.15695#supplemental-information.

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
