# Peer review of "Identifying the critical state of cancers by single-sample Markov flow entropy"

_PeerJ, doi:10.7717/peerj.15695_

## Round 0.1 · original submission · Major Revisions

Your manuscript has now been seen by two reviewers. You will see from their comments below that while they find your work of interest, some major points are raised. We are interested in the possibility of publishing your study, but would like to consider your response to these concerns in the form of a revised manuscript before we make a final decision on publication. We therefore invite you to revise and resubmit your manuscript, taking into account the points raised. Please highlight all changes in the manuscript text file.

Reviewer 1 ·

Basic reporting

The authors develop a new computational method, single-sample Markov flow entropy (SMFE), revealing critical states of complex biological systems. The proposed method has exhibited a good performance for four bulk-sequencing tumor datasets. In addition, the identification of SMFE signaling genes mentioned in the paper helps to discover new prognostic biomarkers and reveal molecular mechanisms related to tumor progression. A few following suggestions are listed for improving the manuscript.
1.In Introduction section, a few related literatures were missed, e.g. PMID: 33956788 and PMID: 33434272, which could contribute to better present the related pioneer works. So, I think that the authors should add these recent advances.
2.In the page 8 of the manuscript, the terms “DEGs” should be “differentially expressed genes (DEGs)”.
3.In the page 9 of the manuscript, the term “DNBs” should be “Dynamic network biomarkers” and supplemented in List of abbreviations section.
4.Third is about the dynamic processes. TCGA seems no information of the stage information.
Finally, the algorithm to extract the marker genes contains no detailed information. An accompanying description of the algorithm in the text would be appreciated

Experimental design

The authors develop a new computational method, single-sample Markov flow entropy (SMFE), revealing critical states of complex biological systems.

Validity of the findings

The proposed method has exhibited a good performance for four bulk-sequencing tumor datasets.

Reviewer 2 ·

Basic reporting

(see additional comments)

Experimental design

(see additional comments)

Validity of the findings

(see additional comments)

Additional comments

Authors reconstruct protein-protein interaction networks based on previously published methods (GNIPLR), and use those networks to define “tipping points” before disease progression. Authors claim that in all datasets considered here, their method sMFE (single-sample Markov flow entropy) is a metric which is indicative of “early warning of the critical transition to disease deterioration” – which is defined as metastatic / distant disease across several cancer types in TCGA.

Authors do two types of analysis: 1) retrospective comparisons of patients in various disease states, which then are fed into 2) prognostic biomarker predictions of early stage cancer survival. The work is generally interesting, and I outline comments below to clarify some intuition behind the analysis choices:

Comments:
1. Lines 43 – 45: This description is overly broad, referring to “complex bioligcal systems” and “fluctuations” – precisely what types of diseases is this in reference to, and what is fluctuating? Gene expression? Micro-environmental conditions?
2. In general, the manuscript would be much improved by precisely defining the following terms in the introduction:
a. Criticality
b. Fluctuations
c. Disease deterioration (what clinical metric?)
3. I also think the manuscript would benefit from an intuitive understanding of why the sMFE metric is expected to have predictive value. Why should sMFE correspond to disease progression? Provide some intuition for the readers.

---

## Round 0.2 · accepted · Accept

Thank you for the detailed response letter and revised version of the manuscript. We are delighted to inform you that your submission has been accepted for publication.

Reviewer 1 ·

Basic reporting

I agree to accept this work.

Experimental design

I agree to accept this work.

Validity of the findings

I agree to accept this work.

Additional comments

I agree to accept this work.

Reviewer 2 ·

Basic reporting

The author's reporting of results are much improved in this new version, with clear definitions for Criticality, Fluctuations, and Disease deterioration, as well as providing a more extended explanation of the intuition drawn from the sMFE metric introduced.

Experimental design

Mathematical modeling approaches are clearly outlined in the methods, and are appropriate for the study question at hand.

Validity of the findings

Authors have satisfactorily addressed my previous concerns/questions. Very nice manuscript.